# Use of Central Nervous System (CNS) Medicines in Aged Care Homes: A Systematic Review and Meta-Analysis

**DOI:** 10.3390/jcm8091292

**Published:** 2019-08-23

**Authors:** Syed Shahzad Hasan, Syed Tabish Razi Zaidi, Jorabar Singh Nirwan, Muhammad Usman Ghori, Farideh Javid, Keivan Ahmadi, Zaheer- Ud-Din Babar

**Affiliations:** 1Department of Pharmacy, School of Applied Sciences, University of Huddersfield, Huddersfield HD1 3DH, UK; 2School of Healthcare, University of Leeds, Leeds LS2 9JT, UK; 3Lincoln Medical School—Universities of Nottingham and Lincoln, College of Science, Lincoln LN6 7TS, UK

**Keywords:** aged care homes, antipsychotics, antidepressants, antiepileptics, antiparkinsons, benzodiazepines, Central Nervous System, psychotropics

## Abstract

Background: Both old age and institutionalization in aged care homes come with a significant risk of developing several long-term mental and neurological disorders, but there has been no definitive meta-analysis of data from studies to determine the pooled estimate of central nervous system (CNS) medicines use in aged care homes. We conducted this systematic review to summarize the use of CNS drugs among aged care home residents. Methods: MEDLINE, EMBASE, CINAHL, Scopus, and International Pharmaceutical Abstracts (IPA) databases were searched (between 1 January 2000 and 31 December 2018) to identify population-based studies that reported the use of CNS medicines in aged care homes. Pooled proportions (with 95% confidence interval), according to study location were calculated. Results: A total of 89 studies reported the use of CNS medicines use in aged care. The pooled estimate of CNS drug use varied according to country (from 20.3% in Ireland to 49.0% in Belgium) and region (from 31.7% in North America to 42.5% in Scandinavia). The overall pooled estimate of psychotropic medicines use was highest in Europe (72.2%, 95% CI, 67.1–77.1%) and lowest in the ANZ region (56.9%, 95% CI, 52.2–61.4%). The pooled estimate of benzodiazepines use varied widely, from 18.9% in North America to 44.8% in Europe. The pooled estimate of antidepressant use from 47 studies was 38.3% (95% CI 35.1% to 41.6%), with the highest proportion in North America (44.9%, 95% CI, 35.3–54.5%). Conclusion: The overall use of CNS drugs varied among countries, with studies from Australia New Zealand reporting the lowest use of CNS drugs. The criteria for prescribing CNS drugs in clinical practice should be evidence-based. The criteria should be used not to prohibit the use of the listed medications but to support the clinical judgement as well as patient safety.

## 1. Introduction

Population ageing is defined as a shift in the age distribution of a population toward older ages [1]. In 2015, adults aged 60 years and older constituted about 12% of the global population (900 million) [1]. Ageing population is projected to increase to 1.4 billion by 2030 (16.5% of the global population) and to 2.1 billion by 2050 (21.5% of the global population) [1]. Parallel to population ageing, the number of aged care homes (also referred to as “nursing homes” or “aged care facilities”) has grown rapidly during the past two decades to partially address the health care needs of older adults [2]. About 12 million older adults needed long-term care facilities in 2012; and this figure is projected to increase to 27 million in 2050 [3].

Old age and institutionalization in aged care homes come with a significant risk of developing several long-term chronic mental health conditions such as cognitive decline, depression, dementia and/or psychosis [4,5]. According to the National Council on Ageing, approximately 92% of elderly adults have at least one chronic disease and approximately 77% have at least two. Dementia is the most common cognitive health issue affecting approximately 47 million adults worldwide with a prediction of a three-fold increase in the number by 2050 [5]. According to the World Health Organization (WHO), over 15% of older adults over the age of 60 suffer from mental illness with depression being a common disorder, affecting nearly 7% of them [5]. An increase in the number of the ageing population could mean the need for more care homes/facilities to provide care for the older population who would need to be institutionalized in such facilities.

Most medicines used to treat central nervous system (CNS) disorders e.g., psychotropics, antipsychotics, antidepressants, benzodiazepines, antiepileptics and antiparkinsons are associated with adverse drug events in older adults [6]. The risk of falling and fall-related fractures have been reported with all of the drugs mentioned above [7,8]. Additionally, these medicines are also associated with cognitive dysfunction and memory impairment [9].

Studies have summarized the use of CNS medicines in individual countries as well as particular region (s) of the world [5,6,10]. However, to the best of our knowledge, there is a lack of a definitive systematic review and meta-analysis of data from studies to determine the global estimate of CNS medicines use in aged care homes. Given the recent technological advances in telecommunication and data management, collaborations between various parts of the world have never been easier. Hence, a comparative study of CNS medicines across various regions and countries of the world will not only provide a useful baseline measure to compare the proportions of CNS medicine use but will also identify the opportunities of collaborative work to optimize the use of CNS medicines. Keeping this in mind, we reviewed the studies reporting the utilization of CNS medicines among residents of aged care homes across the globe.

## 2. Methods

### 2.1. Scope of Review: Eligibility Criteria

The systematic review process was conducted following PRISMA guidelines [11]. To determine the proportions (with 95% CI) of CNS drug use in the aged care homes, a literature search was performed using scientific databases to identify cross-sectional surveys (and prospective or retrospective studies presenting frequencies and percentages) published in full between 1 January 2000 and 31 December 2018. These had to report the use of CNS medicines in older adults (aged ≥ 60 years). Studies were required to recruit participants from aged care settings. The eligibility criteria, which were defined prospectively, are provided in Table 1.

### 2.2. Information Sources

The following databases were searched: MEDLINE (Medical Literature Analysis and Retrieval System Online), EMBASE (Excerpta Medica Database), CINAHL (Cumulative Index of Nursing and Allied Health Literature), Scopus, and IPA (International Pharmaceutical Abstracts). Only articles published between 1 January 2000 and 31 December 2018 were included to present a more recent picture of CNS medicine use in aged care homes. A manual search of the reference lists of the identified articles and recent reviews was conducted to retrieve additional publications.

### 2.3. Search Strategy

The search strategy identified research on CNS medicine use in aged care homes. The medical literature was searched using the following search terms: aged care home, nursing home, residential aged care, or long-term aged care. These were combined using the set operator AND with studies identified with the terms: central nervous system medication or central nervous system medicine or central nervous system drug or CNS medication or CNS medicine or CNS drug or psychotropic, benzodiazepine or antidepressant or antipsychotic, antiparkinson or antiepileptic AND prevalence or utilisation or utilization (see Appendix A for details). The search was limited to original journal articles and observational studies, involving human subjects, older adults and published in the English language. SSH and KA searched the literature, independently to ensure consistency in the results. Titles and abstracts were screened to remove studies that were irrelevant to the aim of the review. The full texts of the remaining studies were then examined to determine eligibility.

Two investigators assessed abstracts independently against 5 criteria: (1) studies conducted to report the proportions of CNS drug use; (2) studies conducted in aged care facilities; (3) cross-sectional surveys or observational (prospective or retrospective) studies; (4) included older adults in care homes; and (5) encompassed patients with a range of CNS related diseases. Full papers of potential studies were independently assessed by the two investigators for their suitability.

### 2.4. Data Extraction

One of the authors extracted data independently to a Microsoft Excel spreadsheet (XP Professional edition; Microsoft, Redmond, Washington, DC, USA) that was verified by the second reviewer. The following data were collected for each study: the name of the first author; country; publication year; study design; duration of the study; number of subjects, and number of females; age of the subjects; any inclusion criteria; types of CNS medicines; description of the CNS related diseases; number of participants; and number of CNS medicine users (see Appendix A).

### 2.5. Assessment of Quality in Included Studies

The lead author independently assessed the quality each of the included studies and discussed their assessments with other two authors to achieve consensus. Each observational study was evaluated against the Scottish Intercollegiate Guidelines Network (SIGN) Methodology Checklist for Observational Studies and ranked in terms of deficiencies [12]. The checklist used to assess the quality of studies based on six standard criteria (domains)was as follows: internal validity; selection of subjects; assessment; confounding; statistical analysis; and overall assessment of the study. The investigators made judgments (using the options, yes, no, can’t say, and doesn’t apply) to indicate whether the studies met the particular criterion.

### 2.6. Data Synthesis and Statistical Analysis

The primary outcome or focus of this review was to report the frequency and proportion of CNS drug use. Pooled estimate or proportion was defined as the number of persons using CNS drugs at a given time divided by the total number of persons who participated in the study. The proportion of individuals using medicines affecting CNS in each study, was combined to give a pooled estimate for all studies. The primary analysis reports an overall use of CNS drugs, whereas subgroup analysis reports the use of individual therapeutic group (e.g., antidepressants, antipsychotics). Heterogeneity between studies was assessed using the I^2^ statistic with a cut-off of 50%, and the χ2 test with a *p*-value < 0.10, was used as the threshold for statistically significant heterogeneity. Subgroup analyses were conducted according to individual country and geographical region. Data were pooled using MetaXL V.5.3 [13] to generate forest plots of pooled estimates with 95% CIs. MetaXL uses untransformed, logit transformed and double arcsine transformed prevalence; for the double arcsine transformation, the default option in MetaXL, was employed in this study [13].

## 3. Results

### 3.1. Characteristics of the Included Studies

The search yielded 17,683 titles from the selected databases. After the removal of duplicate records, and title screening, 313 full-text articles were retained for further evaluation (see Figure 1). In the end, 89 articles met the inclusion criteria and were selected. Detailed characteristics of all included studies are provided in online Appendix A. The majority of studies were conducted in Europe (*n* = 32), Scandinavia (*n* = 23), North America (*n* = 17), and Australia-New Zealand (ANZ) (*n* = 15) with a minority of the studies coming from South East Asia (*n* = 2, Singapore and Malaysia). Most studies were published in the US (*n* = 12) followed by Australia (*n* = 11) and Norway (*n* = 10). There were few studies from Asia, and no studies from Africa, or South and Central America.

### 3.2. Trends in Global Use of CNS Drugs

The CNS drug use in the aged care settings, when data from all 89 separate study populations were pooled, was 35.0% (95% CI 33.1% to 36.8%) (Table 2). The pooled estimate of CNS drugs use in 2001 was 27.5% (95% CI 12.0% to 61.9%) and 31.4% (95% CI 20.5% to 43.5%) in 2018. The pooled estimate of CNS drug use in individual countries is provided in Figure 2, and the pooled estimate of CNS drug use according to geographical study location is provided in Table 2. There was statistically significant heterogeneity between studies in all of these analyses (see Appendix A). The highest percentage of CNS drug use occurred in Scandinavia, parts of Europe (42.5%; 95% CI 39.8% to 45.3%) and the lowest in North America (31.7%; 95% CI 27.8% to 35.7%). Table 3 presents the pooled estimate of overall CNS drug use among aged care home residents by countries. The highest pooled percentage of CNS drug use was in Belgium (49.0%; 95% CI 37.5% to 60.4%) followed by Finland (44.8%; 95% CI 37.9% to 51.8%) and Austria (44.3%; 95% CI 33.6% to 55.2%).

### 3.3. Use of Psychotropic Drugs

The overall pooled estimate of psychotropic drug use from 32 studies was 68.0% (95% CI 65.7% to 70.2%), with the highest percentage in Europe (72.2%) and the lowest in the ANZ region (56.9%) (Table 4). Overall, more than half of the residents of aged care homes were prescribed with psychotropic drugs, as shown in Figure 3, with the highest percentage of 79.7% reported in a study conducted in Finland [14]. The highest pooled estimate (78.0%; 95% CI 74.2% to 81.6%) was in Finland, followed by Belgium (77.2%; 95% CI 72.9% to 81.2%) (Table 5).

### 3.4. Use of Antipsychotic Drugs

Approximately a quarter of residents in aged care homes from 57 studies received treatment with antipsychotic drugs (26.1%; 95% CI 25.1% to 27.2%) (Table 3). The utilisation differed according to geographical regions: from 23.5% (95% CI 20.7% to 26.4%) in ANZ to 27.7% (95% CI 23.9% to 31.6%) in Europe (Table 3 and Figure 3). The highest percentage reported was 45.9% in a study conducted in Austria [15,16]. The highest pooled percentage (45.9%) was in Austria, followed by Finland (37.0%) (Table 4).

### 3.5. Use of Antidepressant Drugs

The pooled estimate of antidepressants use from 47 studies was 38.3% (95% CI 35.1% to 41.6%) (Table 3). The utilisation according to geographical regions were as follows: 44.9% in North America, 41.0% in Scandinavia, 33.5% in Europe, and 30.5% in ANZ (Table 3 and Figure 3). The highest percentage reported was 59.8% in a study conducted in the US [17]. The highest pooled percentage (52.0%) was in the US, followed by Sweden (43.9%) (Table 4).

### 3.6. Use of Benzodiazepines

The pooled estimate of benzodiazepine use from 70 studies was 36.2% (95% CI 32.2% to 40.4%) (Table 3). The percentage use of benzodiazepines varied widely from 18.9% in North America (*n* = 9) to 44.8% in Europe (*n* = 14) (Table 3 amd Figure 3). The highest percentage reported was 64.2% in a study conducted in the Netherlands [18]. The highest pooled percentage (54.1%) was in Belgium, followed by France (48.4%) (Table 4).

### 3.7. Use of Antiepileptic and Antiparkinson Drugs

The pooled estimate of antiepileptic drug use from 17 studies was 10.8% (95% CI 9.1% to 12.6%) (Table 3). The use did not vary greatly among the countries with a range of 9.8% in Norway to 13.5% in Italy. Only four studies met the criteria that reported the proportion of antiparkinson drugs among aged care home residents. The pooled estimate of antiparkinson drug use was 6.5% (95% CI 5.2% to 8.0%) (Table 3).

## 4. Discussion

This systematic review with meta-analysis assembled data from 89 population-based studies, reporting the proportion of CNS drug use in the aged care homes. We found a relatively high prevalence of CNS drug use in aged care homes in most regions, especially Scandinavia, Europe and North America. The proportion of CNS drugs use varied, from 31.7% to over 42.5%, according to the geographical location of the population under study, with an overall proportion of CNS drug use of 35.0%. A higher proportion of residents in aged care homes in European and Scandinavian regions were using psychotropic medicines while the use of antipsychotic drugs did not vary greatly across geographical regions. There were no studies from African or South American countries, and it is possible that studies identified in this search were conducted in countries where the culture of having elderly adults in nursing homes is acceptable, such as in the United States of America (US), Europe, ANZ and some Asian countries; whereas, in developing regions, older adults were cared for by family members.

In this study, the overall pooled estimate of psychotropic drug use from 32 studies was 68.0%. This high percentage of psychotropic medicines use in aged care homes compared with the normal population is of great importance from both a policy-making point of view and a clinical point of view. CNS drug use is justified, as these medicines treat several mental disorders, including depression, psychiatric and mood disorders. Hence, one could argue that from policy-making point of view they should be relatively easy to prescribe and dispense in order to treat depression, aggression, mood disorders, etc., to ensure the well-being of aged care homes’ residents. From a clinical point of view these medicines may produce adverse effects, particularly in long-term therapy and in older frail adults who are taking a number of other medications for their chronic conditions. Moreover, with ageing comes pharmacokinetic changes, which would mean the need for dose adjustment to avoid toxicity, monitoring drug drug interactions, monitoring drug disease interactions, etc. [19].

### 4.1. Trend in the Global Use of CNS Drugs

Scandinavia and Europe (Belgium and Austria) had the highest pooled proportion of any CNS medicine use. Increased prevalence of CNS drug use and high-volume prescriptions of CNS drugs such as benzodiazepines and antiepileptic drugs have been reported in Norway, Austria and Belgium [19,20,21,22,23].

Studies have suggested a variety of reasons for the persistent use of CNS drugs in aged care homes. Management of neuropsychiatric symptoms—behavioural symptoms, for instance, has been associated with frequent psychotropic drugs prescription [21,22]. Prescribers’ behaviour has been suggested as another reason for the high prevalence of benzodiazepines use among aged care home residents [20]. Lack of communication between general practitioners (GPs) and nurses has contributed towards high prevalence of CNS drug use in Belgian aged care homes [23]. In an attempt to broadly classify the reasons for the high prevalence of CNS drug prescription in aged care homes, we can categorize them into (a) patients characteristics/needs and (b) policy implementation.

It has been reported that a high prevalence of certain conditions such as dementia is associated with high prevalence of CNS drug use [21,24]. To control behavioural symptoms and pain one would think that it is in the patient’s interest—as well as in the interest of other residents of aged care homes—to ensure patients receive their medication. However, it has also been reported that in the process of CNS drug prescription and dispensing, sometimes, guidelines and policies of managing medicines in aged care homes are not fully implemented [20,24]. Healthcare professionals could be mentally juggling between a patient’s desires and that patient’s interest, especially in dire situations where there is an immediate need to intervene to alleviate distress and pain in elderly frail patients/residents of aged care homes.

Nonetheless there are also some good examples of commendable practices of lower than-average CNS drug use. Australia New Zealand and the United Kingdom, for example, had the lowest pooled proportion of any CNS medicine use in aged care homes. From clinical point of view, these countries have adapted a more proactive approach to behavioural symptoms and pain control by ensuring timely dose tapering without re-emergence of symptoms. Moreover, they have introduced a multidisciplinary approach to patient care by including different healthcare professionals to answer patients’ needs through both pharmacological and non-pharmacological interventions [25]. Non-pharmacological intervention would automatically reduce the volume and frequency of pharmacological treatment modalities.

From a policy implementation point of view, detailed guidelines and pragmatic protocols have been devised to protect both patients/residents as well as healthcare professionals [26]. Such guidelines would serve as a legal framework not to only oblige the healthcare professionals to follow the rules; but they would also, protect everyone, especially securing patients/residents of aged care homes’ safety.

### 4.2. Benzodiazepines

Benzodiazepines are one of the most commonly used CNS drugs for treating conditions like insomnia and anxiety; however, studies have reported that for most residents, no appropriate indication was documented [6,27,28]. Most studies carried out on the effect of benzodiazepines on the above two indications were conducted on young adults and limited data are available on long-term effects and efficacy of these medications [29,30,31,32]. While short-term use of benzodiazepines might be safe in young adults, its short-term use in elderly patients can have serious adverse effects, such as an increase in falls and fractures, as well as slowed reaction time, leading to motor vehicle accidents [33,34]. Additionally, benzodiazepines use in the elderly could cause dependence, withdrawal effects and long-lasting cognitive decline and can lead to dementia due to the slow rate of metabolism in older adults [35,36].

Among the benzodiazepines, short-acting agents such as lormetazepam, lorazepam and oxazepam were most frequently used by the residents [6,14,37,38,39,40,41,42]. It should be noted that short-acting drugs induce a greater risk of withdrawal symptoms, as the body has less time to adapt to working without them. Duplicate therapy (e.g., patients using two benzodiazepines), which is of concern regarding benzodiazepine prescribing, occurred among 13.6% and 9.7% of French [38] and Belgian [6] residents, respectively; however, this was nearly four times lower among Italian (3.7%) [43] and Spanish (1.9%) [41] residents. These results may have far-reaching implications in clinical practice and research related to the management of CNS disorders in older adults. Being aware of individuals most at risk will allow clinicians and researchers to direct their efforts in developing guidelines more suited to high-risk groups (e.g., those having a greater risk of withdrawal symptoms).

### 4.3. Antidepressants

Depression is one of the most common disorders in the elderly population, and has historically been undiagnosed and untreated [5]. As the number of older persons is increasing the number of elderly patients suffering from mental and neurological disorders including depression is also increasing [5]. Depression is associated with functional decline, which could also negatively affect other co-morbidities such as stroke, and it could eventually lead to death due to suicide [44]. This requires a higher degree of care and the use of care facilities, which may lead to the higher use of antidepressants. In choosing an appropriate antidepressant drug for an elderly patient the side effect profile and lowest risk of drug interaction must be considered. According to the NICE guidelines (CG90) updated in 2016, the first line drug treatments are the selective serotonin reuptake inhibitors (SSRIs), which are relatively safe in the elderly [45].

The safer SSRIs for use in elderly patients are sertraline, citalopram and escitalopram [46,47,48]. If tricyclic antidepressants (TCAs), which are second line drug treatments, have to be given, then desipramine and nortriptyline are the best choices, as they are less anticholinergic [47]. Other tricyclic drugs should be avoided due to worsening effect on dementia, Parkinson’s disease, cardiovascular problems and postural hypotension which would increase the risk of fall and fractures in elderly patients. According to a Cochrane review, both SSRIs and tricyclic antidepressants are equally effective in the elderly, although SSRIs are better tolerated with fewer adverse effects [49]. It should also be taken into consideration that older patients take longer to show a response to treatment [49,50].

There was a major change in the pattern of prescribing in aged care homes from the late 1990s, when TCAs, particularly amitriptyline and nortriptyline, were largely replaced by SSRIs, especially sertraline, citalopram and escitalopram. The latter became the most frequently used antidepressants in aged care homes in most countries, including Belgium [6,27,51], Finland [14], Norway [10,52,53], Sweden [54,55,56,57], Austria [15], Slovenia [42], US [37,58,59], and Singapore [60], as well as ANZ [61,62,63,64]. However, mirtazapine, which is an atypical antidepressant, was most frequently prescribed in other studies conducted in the US and Sweden [17,39]. Meanwhile in Germany, TCA use was favoured [16]. A more worrying analysis showed that more than one type of antidepressant was prescribed concomitantly in 1% to 8% of aged care residents [6,14,17,43,62,65]. Considering other comorbidities that elderly patients have, there is a high chance of drug-drug interactions, and therefore, this proportion could be a target for improvement. Lack of documentation regarding indications was also an issue and in Swedish aged care homes more than half (65.0%) of the antidepressant users had no documented indication for treatment [55]. However, in studies conducted in Norway and the United Kingdom (UK), aged care homes reported this percentage to be less than 15.4% and 42.5%, respectively [17,66]. It is worth noting that under-treatment of depression was also documented in some studies, in which at least one quarter of depressed residents did not receive any antidepressant medicine [17,55,66]. It was also not clear whether patients received any non-pharmacological cognitive therapy in addition to or prior to the drug therapy. These findings reinforce the need for rational guidelines to optimize therapy with antidepressants and medicine management of depression among residents in aged care homes.

### 4.4. Antipsychotics

In recent years, the use of typical or conventional antipsychotics with a strong antagonistic effect on dopamine D2 receptors have been avoided in elderly patients due to their severe adverse effects and limited tolerability [67,68]. On the other hand, atypical antipsychotic medicines with high affinity for both serotonergic and dopaminergic receptors and other receptors, have been widely used in recent years due to their effectiveness against negative symptoms, their higher efficacy and binding profile and, more importantly, their limited extrapyramidal side effects in patients diagnosed with schizophrenia, mania and acute psychosis reactions and other psychotic disorders [69,70,71,72]. It seems that atypical antipsychotics reduce dopamine transmission by acting on the mesocortical pathway which leads to anti-psychotic effects, while increasing dopamine transmission in the frontal cortex and nigrostriatal pathway [72,73]. In addition, the transient binding and dissociation of atypical anti-psychotics to D2 receptors would lead to a normal transmission of dopamine, which would constitute a more suitable option particularly for elderly patients [72,73]. However, even the atypical anti-psychotics induce some side effects in elderly populations.

Knowledge of the binding profile and adjusting the dose of antipsychotics play an essential role in the effects they mediate or the symptoms they alleviate. For example, risperidone equally blocks D2 and 5-HT2 receptors; however, at higher doses of more than 2 mg/day, it has a higher affinity for D2 receptors, which may facilitate the onset of extrapyramidal side effects in elderly patients [73,74,75]. Ziprasidone blocks more 5-HT2 receptors than D2 receptors and is more effective in reducing psychotic symptoms and better tolerated than typical anti-psychotics such as haloperidol, particularly in movement disorders [74,76]. Interestingly, its weak anticholinergic effect explains its more favourable cognitive profile than typical drugs in elderly patients. The lack of cognitive impairment has also been reported by olanzapine which stimulates the release of acetylcholine by blocking 5-HT6 receptors [77].

The main indications for prescribing antipsychotics to residents in aged care homes were aggression, agitation, and restlessness associated with dementia [6,42,78,79]. However, a documented indication for antipsychotic treatment was absent for 48% of antipsychotic users in a study conducted in Sweden [55]. Briesacher et al. reviewed the indications for antipsychotic prescribing in American nursing homes during the year 2000–2001 and found that there was no appropriate indication in a lower proportion (23.4%) of antipsychotic users [79]. In another study in the US conducted in 2004, the proportion was almost two-fold higher (40.0%) [28]. In Irish nursing homes, 51.0% of antipsychotic users were deemed to be “receiving antipsychotics inappropriately” [80]. Furthermore, a UK study reported that more than half (54.0%) of the antipsychotic prescriptions in aged care homes were “inappropriate” [81], while two other studies (in France and Belgium) revealed that more than two thirds (66.5 to 98.0%) of antipsychotic users were found to have “potentially inappropriate prescribing” [78,82]. This was ascribed to a high prevalence of chronic use, pointing to a lack of reassessment [78]. Combination antipsychotic therapy, which is problematic, was found in 3.5–15% of residents [6,43,65,83].

### 4.5. Antiepileptic and Antiparkinson Medicines

Our meta-analysis raises concerns regarding the undocumented indication for the use of anti-epileptic drugs (AEDs) in aged care home residents. This was illustrated in Germany, where the indication for AED use was missing in one quarter of residents receiving AEDs [84]. This figure was higher in Austria, where an indication for AED use could not be ascertained in 39% of residents [85]. Compared to studies in the US and European countries, which reported percentages of less than 50%., more than three quarters of residents in Singaporean homes received AEDs without proper documentation of a therapeutic indication [60].

In prescribing anti-epileptic drugs to elderly patients, clinical practitioners should consider the adverse effects and drug-drug interactions. The majority of anti-epileptic drugs interact with other medications as well as hepatic enzymes and plasma proteins. Newer anti-epileptic drugs such as gabapentin, tiagabine and levetiracetam have minimum interaction and, therefore, are safer for elderly patients [86,87]. Although the International League Against Epilepsy Guidelines recommend newer AEDs, such as lamotrigine and gabapentin, for elderly adults, most AEDs used in the studies in this review were contrary to evidence-based recommendations. Older AEDs, such as phenobarbital were used most frequently in aged care homes in Italy [88,89], while carbamazepine was most frequently used in Norwegian (21.4%) [19], German (37.0%) [84], and Swedish homes (32.0%) [90]. Gabapentin was the most frequently used AED (37.0%) in Austrian aged care homes [85].

It is well-known that AEDs are frequently implicated in drug drug interactions, which can be attributed to their complex pharmacokinetic properties. Therefore, it is not surprising that a recent study (2016) from Norway reported that drug-drug interactions were observed in nearly half (45%) of AED users in aged care homes [19]. Efforts should be made to implement rational therapy guidelines and optimize drug management of neurological and psychiatric disorders among of aged care home residents as they represent the frailest segment of the geriatric population. Additional studies are needed to determine the use of CNS medications, particularly in Asian countries.

Only four studies reporting the use of antiparkinson drugs were included in this study showing a pooled estimate of 6.5% (95% CI 5.25 to 8.0%). Antiparkinson drugs work very well for the majority of patients at the beginning of their therapy; however, this may change over time and side effects, such as wearing off and dyskinesia, hallucinations and delusions, orthostatic hypotension, and the occurrence of impulsive and compulsive behaviour. Upon initiation of the treatment for Parkinson’s disease in elderly patients the cognitive issues and comorbidities should be considered. The initial treatments are with dopamine agonists and levodopa [91]. The optimum dose for each patient might vary and should be identified as a dose that works the best for each elderly patient. The optimum dose of levodopa tends to remain constant over the years although more frequent doses might be required to counteract the levodopa wearing-off effects but not higher individual doses [92]. The ceiling dose should not be exceeded, as beyond this there is no further benefit. Clinicians can escalate the dose to capture the most benefit for any elderly patient [93].

## 5. Conclusions

The present meta-analysis has demonstrated that a significant proportion of older people residing in care homes consume CNS medicines with psychotropic medicines being the most commonly used medicines. The overall use of CNS drugs varied among countries, with studies from Australia New Zealand reporting the lowest use of CNS drugs. Elderly patients in care homes are among the most vulnerable members in society, and rely on care home staff for their everyday needs. There is always a risk in the use of medications in the elderly population due to the changes in the metabolism, which alter the pharmacodynamics and pharmacokinetics of drugs, increasing the risk of interactions particularly due to the frequency of polypharmacy in these patients. Therefore, careful balance between treating safely and adequately should be considered to avoid further comorbidity and risk of death in older populations. Careful monitoring is needed for adverse effects and attention should be given to some medications which may also cause harm. The criteria for prescribing these medications in clinical practice should be evidence-based. The criteria should be used not to prohibit the use of the listed medications but to support the clinical judgement. The quality of evidence, potential for harm and availability of safer alternatives should be considered. Educating the patients can play an important first step in tapering protocols for medications such as benzodiazepines for safe and comfortable discontinuation of the medication. Future studies should focus on strategies for the development of specific guidelines for aged care home residents receiving CNS drugs.

## Figures and Tables

**Figure 1 jcm-08-01292-f001:**
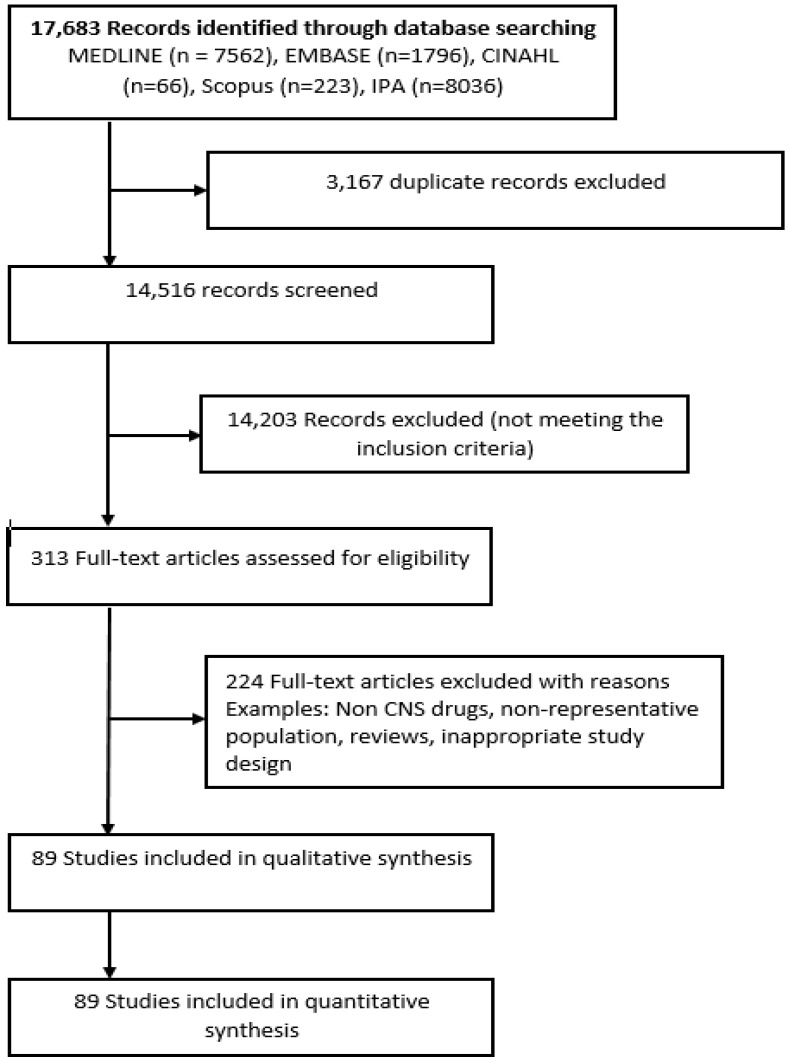
Study selection process (PRISMA).

**Figure 2 jcm-08-01292-f002:**
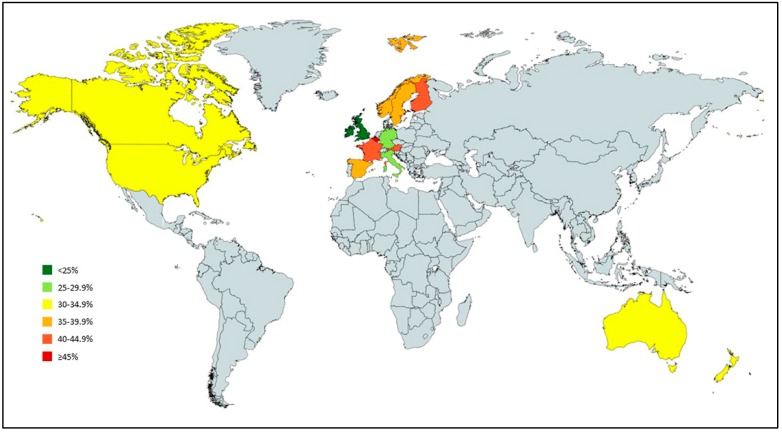
Overall CNS drug use among care homes residents (countries with at least 2 studies).

**Figure 3 jcm-08-01292-f003:**
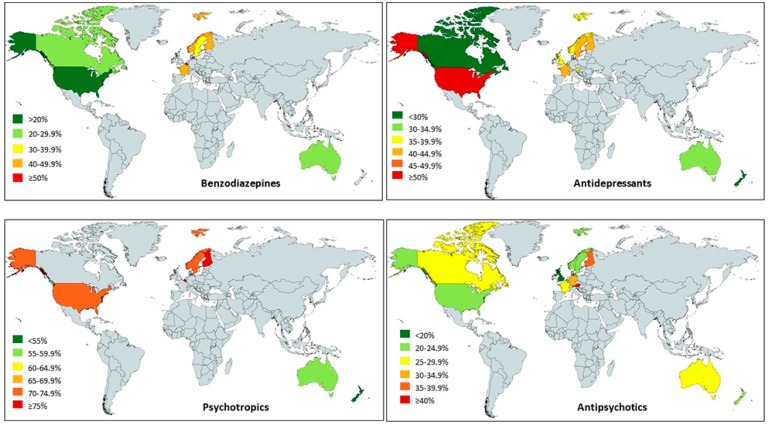
Use of psychotropic, antipsychotic, antidepressant and benzodiazepine drugs among care home residents (countries with at least 2 studies were included).

**Table 1 jcm-08-01292-t001:** Eligibility criteria.

Cross-Sectional Surveys or Observational (Prospective or Retrospective) Studies
Recruited adults (>90% of participants aged ≥60 years)
Participants recruited from the aged care homes
Reported frequency and/ or percentage of CNS drug use (total sample and number of users)
Sample size of ≥50 participants
Broad definition of CNS drugs including psychotropic, antipsychotic, antidepressant, benzodiazepines, antiparkinson, and antiepileptic medicines

**Table 2 jcm-08-01292-t002:** Pooled proportion of any CNS medicine use according to geographical location.

Region	Studies (n)	Pooled Proportion (%)	95% CI (%)	I^2^ (%)	*p*-Value for I^2^
Scandinavia	23	42.5	39.8–45.3	99.4	<0.001
Europe	32	41.2	36.0–46.6	99.8	<0.001
South East Asia	2	38.2	13.8–65.8	97.4	<0.001
ANZ	15	34.3	28.1–40.8	99.7	<0.001
North America	17	31.7	27.8–35.7	99.9	<0.001
All studies	89	35.0	33.1–36.8	99.9	<0.001

ANZ = Australia and New Zealand.

**Table 3 jcm-08-01292-t003:** Overall CNS drug use among care home residents by countries.

Country	No. Studies	Pooled Proportion (%)	95% CI (%)
Belgium	4	49.0	37.5–60.4
Finland	4	44.8	37.9–51.8
Austria	3	44.3	33.6–55.2
France	4	41.3	30.1–52.9
Sweden	9	38.9	30.0–48.2
Norway	10	38.7	31.6–46.0
Spain	2	37.0	19.5–56.3
US	12	31.7	26.9–36.7
Australia	11	31.7	24.6–39.3
New Zealand	4	31.3	15.1–49.9
Canada	5	30.0	23.3–37.3
Italy	5	27.7	9.5–50.3
Germany	3	25.1	14.1–38.0
UK	5	22.3	16.5–28.7
Ireland	2	20.3	14.7–26.6

Countries (Singapore, Malaysia, Switzerland, Slovenia, and the Netherlands) with only one study were excluded.

**Table 4 jcm-08-01292-t004:** CNS medicine use according to therapeutic classification and geographic classification.

Drug	Studies (n)	Pooled Proportion (%)	95% CI (%)	I^2^ (%)	*p*-Value for I^2^
Psychotropic drugs	32	68.0	65.7–70.2	98.5	<0.001
Europe	10	72.2	67.1–77.1	99.0	0.001
Scandinavia	10	71.6	69.2–74.0	91.0	0.001
North America	3	69.3	65.0–73.6	98.0	0.001
ANZ	9	56.9	52.2–61.4	98.0	0.001
Antipsychotics	57	26.1	25.1–27.2	99.6	<0.001
Europe	18	27.7	23.9–31.6	99.0	<0.001
Scandinavia	14	26.3	22.8–30.0	98.0	<0.001
North America	15	24.6	22.7–26.6	100	<0.001
ANZ	10	23.5	20.7–26.4	96.0	<0.001
Antidepressants	47	38.3	35.1–41.6	99.7	<0.001
North America	10	44.9	35.3–54.5	100.0	<0.001
Scandinavia	18	41.0	39.2–42.9	93.0	0.001
Europe	13	33.5	29.3–37.7	99.0	<0.001
ANZ	6	30.5	23.6–37.7	96.0	0.001
Benzodiazepines	70	36.2	32.2–40.4	99.8	<0.001
Europe	14	44.8	38.4–51.2	100	<0.001
Scandinavia	18	42.3	37.8–46.8	99.0	<0.001
ANZ	8	29.4	21.1–38.2	99.0	<0.001
North America	9	18.9	13.5–24.8	99.0	<0.001
Antiepileptics	17	10.8	9.1–12.6	98.4	<0.001
Antiparkinsons	4	6.5	5.2–8.0	79.0	0.003

Note: ANZ = Australia and New Zealand; North America includes US and Canada.

**Table 5 jcm-08-01292-t005:** Use of different CNS drugs by residents of care homes by country.

Country	No. Studies	Drug Class	Pooled Proportion (%)	95% CI (%)
Australia	5	Psychotropic	60.2	54.5–65.8
	7	Antipsychotic	24.8	21.3–28.5
	5	Antidepressant	30.6	21.8–40.1
	6	Benzodiazepine	27.4	17.4–38.7
Austria	2	Antipsychotic	45.9	44.3–47.5
	2	Antidepressant	36.8	35.3–38.4
	2	Benzodiazepine	35.0	33.5–36.5
Belgium	2	Psychotropic	77.2	72.9–81.2
	2	Antipsychotic	24.2	10.6–41.0
	3	Antidepressant	40.3	38.8–41.8
	3	Benzodiazepine	54.1	51.8–56.4
Canada	5	Antipsychotic	28.6	24.5–32.9
	3	Antidepressant	27.7	8.9–51.2
	4	Benzodiazepine	23.9	12.3–37.7
Finland	2	Psychotropic	78.0	74.2–81.6
	6	Antipsychotic	37.0	31.9–42.2
	4	Antidepressant	43.3	41.2–45.5
	4	Benzodiazepine	40.8	26.6–55.8
	3	Antiepileptic	10.8	6.5–15.9
France	3	Antipsychotic	26.0	23.8–28.3
	2	Antidepressant	43.3	42.2–44.5
	2	Benzodiazepine	48.4	38.4–58.4
Germany	2	Antipsychotic	31.6	25.0–38.7
Italy	3	Antiepileptic	13.5	3.5–28.0
New Zealand	4	Psychotropic	52.5	40.4–64.4
	3	Antipsychotic	20.3	16.6–24.4
	2	Antidepressant	25.4	16.5–35.5
	2	Benzodiazepine	35.1	32.6–37.7
Norway	8	Psychotropic	70.4	68.0–72.8
	10	Antipsychotic	23.5	22.5–24.5
	12	Antidepressant	38.7	35.5–42.0
	10	Benzodiazepine	44.8	39.2–50.4
	3	Antiepileptic	9.8	7.2–12.7
Spain	2	Benzodiazepine	41.4	32.2–50.9
Sweden	4	Psychotropic	70.1	67.7–72.5
	5	Antipsychotic	20.3	16.4–24.7
	8	Antidepressant	43.9	41.2–46.6
	4	Benzodiazepine	37.1	34.0–40.3
UK	5	Antipsychotic	18.9	17.5–20.3
	2	Antidepressant	35.3	30.1–40.7
US	2	Psychotropic	71.2	68.1–74.1
	10	Antipsychotic	22.8	20.7–24.9
	7	Antidepressant	52.0	43.2–60.7
	5	Benzodiazepine	14.7	8.8–21.8

Note: Countries with a minimum of 2 studies were included.

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
