# Peer review of "Use of Central Nervous System (CNS) Medicines in Aged Care Homes: A Systematic Review and Meta-Analysis"

_jcm, 2019, doi:10.3390/jcm8091292_

Round 1

Reviewer 1 Report

This review is for the submitted manuscript titled “Use of central nervous system (CNS) medicines in  aged care homes: A systematic review and meta-4 analysis”. The manuscript evaluated 89 publications from multiple countries to compare the rates of CNS medicines used in aged care home settings. Although some of the findings may be of interest, the report generally lacks rigor and is a bit more like a school assignment that a strong manuscript for publications. 

Major concerns: 

1) For a valid comparison, similarities between studies must be reported such as type of elder care and average age of patients in each study.

2) 89 publications across multiple countries is a rather small number of studies.

3) Not enough data was provided to support conclusion “Countries with a higher than average use of CNS medicines may benefit from collaborating with countries where such use is below average in curtailing the potentially inappropriate use of CNS medicines.”

Reviewer 2 Report

The systematic review and meta-analysis study determined the pooled estimate of central nervous system medications use among aged care homes residents from total of 89 studies.  The authors reported a varied use of CNS meds among counties and regions.  The study adds knowledge to the literature regarding current trend of use of CNS medications in aged care homes residents. The paper is well written. Authors have mentioned the limitations of the study.

I have few comments to make:

1.       Spelling errors: page 2, line 53 (‘number’ spelled as ‘umber’)

2.       Line 56:  I would avoid using ‘psychotropics’ as it’s a general term for all other class of psychiatric drugs.

3.       Not sure why the CNS drugs ‘stimulants’ were not included in the study

4.       Search terms included CNS drug class (for eg. Antidepressant) but not individual medication (for eg. Fluoxetine). It is possible to miss studies which might have focused on individual CNS drug rather than a class.

5.       Explanation for why search was limited to only journal articles.

6.       Explanation for why only articles published in English was included, when aim of the study is to have global estimates of the CNS drug use.

7.       If there were disagreements during the assessment of quality of the studies, how it was resolved.

8.       Discussion section needs to be more focused on the result findings.

Round 2

Reviewer 1 Report

I do not believe I had access (or I missed it) to the supplementary tables on the original submission. Given the number of participants in each of the 89 studies, I significantly change my view on this manuscripts rigor and appreciate the authors pointing this out. 

Still 2 comments: 

1) can the authors include how many care centers were assessed in each manuscript (in table s2) 

2) regarding my initial comment "Not enough data was provided to support conclusion “Countries with a higher than average use of CNS medicines may benefit from collaborating with countries where such use is below average in curtailing the potentially inappropriate use of CNS medicines." 

I still find that this statement should be changed. No data provided suggests that collaboration between countries will solve the problem of potentially inappropriate use of CNS medications.  
